# Healthcare Professionals’ Perceptions of Pre-, Peri-, and Postoperative Virtual Reality Immersion in Elderly Patients

**DOI:** 10.3390/healthcare13060669

**Published:** 2025-03-19

**Authors:** Kristian Hermander, Pether Jildenstål, Sofia Erestam, Peter Dahm, Sophie Lindgren, Joakim Strömberg, Carina Sjöberg

**Affiliations:** 1Institute of Health and Care Sciences, Sahlgrenska Academy, University of Gothenburg, 405 30 Gothenburg, Sweden; kristian.hermander@vgregion.se (K.H.); pether.jildenstal@vgregion.se (P.J.); sofia.erestam@gu.se (S.E.); 2Department of Anaesthesiology, Surgery and Intensive Care, Sahlgrenska University Hospital, 413 45 Gothenburg, Sweden; peter.dahm@vgregion.se; 3Department of Health Sciences, Lund University, 22100 Lund, Sweden; 4Faculty of Nursing and Health Sciences, Nord University, 8026 Bodø, Norway; 5Department of Anesthesiology and Intensive Care, Skane University Hospital, 222 42 Lund, Sweden; 6Department of Anesthesiology and Intensive Care, School of Medical Science, Örebro University Hospital, 701 85 Örebro, Sweden; 7Department of Anaesthesiology and Intensive Care Medicine, Institute of Clinical Sciences, Sahlgrenska Academy, University of Gothenburg, 405 30 Gothenburg, Sweden; sophie.lindgren@vgregion.se; 8Department of Hand Surgery, Institute of Clinical Sciences, Sahlgrenska Academy, University of Gothenburg, 405 30 Gothenburg, Sweden; joakim.stromberg@vgregion.se; 9Department of Surgery and Orthopaedics, Alingsås Lasarett, 441 33 Alingsås, Sweden

**Keywords:** aged, anxiety, healthcare professionals, perioperative, regional anesthesia stress, surgery, virtual reality

## Abstract

**Background/Objectives:** There is a lack of research examining healthcare professionals’ perspectives regarding the potential of non-pharmacological solutions such as immersive virtual reality (VR). The aim of this study was to investigate opportunities and challenges related to the application of immersive virtual reality (VR) technology in patients aged 65 and older undergoing surgery with regional anesthesia and sedation. **Method:** A qualitative, multicenter study was conducted in the spring of 2024, involving semi-structured interviews with 17 healthcare professionals. **Result:** A qualitative content analysis of the interviews identified the main theme “Healthcare professionals’ openness to opportunities for this technology for elderly patients”, with the subthemes and themes “elderly patients”, which included the participants’ attitudes towards elderly patients; “virtual reality”, which concerned opportunities, barriers, and risks; and “sustainable healthcare”, which comprised the participants’ thoughts about its impact on sustainable development. **Conclusions:** The participants suggested potential areas of use for VR during the perioperative period but also identified limitations and risks. They suggested VR was likely to have a positive impact on sustainable healthcare, as well as economic advantages. For its successful implementation, the equipment must be safe. There also needs to be a clear division of responsibilities for it to be functional and suitable for its users. Strategies such as nudging can be used to facilitate its implementation.

## 1. Introduction

Global demographic trends are characterized by an increasing population over 65 years of age, especially in Europe [1]. Consequently, the prevalence of chronic diseases and comorbidities has also increased, leading to a greater demand for various types of surgical interventions [2]. This trend presents significant challenges for the perioperative care context, necessitating enhanced flexibility and adaptation to meet the unique needs of an older patient population [3,4,5]. The over-65 population experiences age-related physiological changes that affect organ systems, with both short-term risks and long-term adverse outcomes [4]. It is common for elderly patients to have existing cognitive impairment and comorbidities, such as diabetes, which may predispose this age group to a delayed neurocognitive recovery [3,5]. As a result of this disease burden, this patient group is often treated with a variety of drugs and pharmacokinetics, and the pharmacodynamics complicate matters further [6]. When older patients undergo surgery, regional anesthesia (RA) is a common anesthetic strategy, which is combined with intraoperative propofol sedation and potentially supplemented with analgesics when required.

It is well known that the intraoperative use of sedatives can result in a deeper level of sedation than intended, occasionally reaching levels comparable to those with general anesthesia (GA) [7]. This increases the risk of complications, such as pulmonary complications (e.g., aspiration) and cardiovascular complications (e.g., hypotension, frailty, pressure ulcers, and concerns related to polypharmacy) [3,6]. A cohort study by Sprung [8] demonstrated that older adults undergoing surgery with regional anesthesia (RA) experienced a decline in global cognitive function that was similar to that in patients receiving general anesthesia (GA). Additionally, a randomized multicenter study conducted in Australia, China, and the US, comprising patients aged 70 years and older undergoing major surgery, highlighted the significance of anesthesia depth [9]. This study found that for patients whose anesthesia was guided by bispectral index (BIS) monitoring, with a target BIS of 50 compared to 35, the BIS 50 group exhibited significantly better cognitive function one year post-surgery [9].

In a technology-dense environment, patient participation plays a crucial role for older adults, as the pre-, per-, and postoperative context and multimodal analgesia, including non-pharmacological methods, should be considered [10]. Virtual reality (VR) has been investigated for its sedative and analgesic properties, as it immerses patients in an artificial, interactive environment [11,12]. VR engages multiple senses simultaneously, allowing patients to hear and experience stimuli that correspond to visual images. VR allows for interaction with the virtual environment and real-time responses [13,14]. Importantly, the use of VR does not hinder communication with individuals in the surrounding environment [15]. Recent advancements in VR technology have been applied to enhancing patient participation, autonomy, and responsibility in perioperative care, promoting person- and child-centered approaches. Moreover, a patient’s experience during the pre-, peri-, and postoperative phases can be positively influenced by offering VR as an adjunct to regional anesthesia to reduce pain and anxiety [15,16,17,18]. A systematic review by Ioannou [19] indicated that VR effectively reduces anxiety, depression, fatigue, and pain in various contexts.

However, there is a lack of studies that have investigated the potential for non-pharmacological solutions such as VR in individuals aged 65 and older. The implementation of VR technology in perioperative care would necessitate modifications to existing workflows, potentially affecting patients and surgical teams. Furthermore, the implementation process must be designed according to scientific principles, where Roger’s theory of the diffusion of an innovation in a clinical environment can be used. According to Rogers [20], characteristics such as an individual’s ability to learn to use an innovation, the observed effects of the innovation itself, the motivation and ability of users, and an assessment of the implications for the organization influence whether an innovation is adopted [20]. Therefore, understanding the perspectives of healthcare professionals is essential, as well as the perceived advantages, disadvantages, and feasibility of utilizing VR in this context. The aim of this study was to investigate the opportunities and challenges related to the application of immersive virtual reality (VR) technology in patients aged 65 and older undergoing surgery with regional anesthesia and sedation.

## 2. Materials and Methods

### 2.1. Design

This study employed a qualitative inductive design. Data were gathered through semi-structured individual interviews. A qualitative content analysis was used to identify patterns, variations, differences, and similarities in the informants’ perceptions [21].

### 2.2. The Setting and Participants

The study participants (n = 17) were certified registered nurse anesthetists (CRNAs), certified perioperative nurses (CNORs), radiographers, and critical care nurses (CCNs) working in units in which we planned to conduct further studies within this project. They were recruited from three different hospitals, including their surgical departments (one at a university hospital and two at regional hospitals). The procedures were performed using different types of regional anesthesia. These procedures included open inguinal hernia surgery with implants with combined regional block and infiltration anesthesia; hybrid and interventional procedures with infiltration anesthesia; and orthopedic arm/hand surgery with regional block and infiltration anesthesia. Purposeful sampling was used to ensure recruitment from various professional categories with experience caring for patients over 65 years of age during the perioperative care process. The sociodemographic and professional characteristics of the participants are presented in Table 1.

The nurse manager of the department was contacted to facilitate the authors’ ability to gather on-site information for this study. Individuals who met the inclusion criteria for this study received oral and written information about this study. They were asked to review the information and were able to ask questions before deciding whether to participate in this study. They were then contacted again, and face-to-face interviews were arranged with the participants. Their written consent was obtained before the interviews were conducted.

### 2.3. Data Collection

Individual semi-structured interviews were held using an interview guide. At the same time as the interviews were conducted, the interviewees had the opportunity to familiarize themselves with the glasses, the hardware.

The interview questions were about their knowledge of, attitude towards, and opportunities to use and obstacles to using VR in the perioperative care process. Practical issues were also addressed, such as aseptic techniques and the compatibility of the VR glasses with hearing aids and sleep depth monitoring. The other questions concerned their approach to VR from a proxy perspective, an economic perspective, and a sustainable development perspective.

A total of 17 interviews took place between February and June 2024. The participants were interviewed at their workplace in private during work hours. Three pilot interviews were conducted by the author who conducted the interviews (K.H.). The author S.E. participated in these interviews to ensure that they were conducted in the same way. The pilot interviews were included in the results. The interviews were then conducted by K.H. All interviews were audio-recorded using an external device and professionally transcribed verbatim. Before the start of the interviews, the interviewer read a pre-defined definition of virtual reality aloud and showed them a prototype. All of the interviews started with the opening question “How do you think the glasses can be used here in anesthesia and surgical care?”. The interviews were then conducted using open-ended questions and supplementary questions, thus ensuring that the participants could use their own words and speak freely about the subject. Before the end of each interview, a summary was made, where the participants were able to determine whether the interviewer’s interpretation was correct. The duration of the interviews ranged from 14 to 28 min. The authors agreed that data saturation was reached when the interviewees repeated remarks that had been made in previous interviews. The sample size was sufficient for this homogenous study population given the narrow research topic [22].

### 2.4. The Data Analysis

The data analysis was performed systematically in accordance with a qualitative content analysis [23]. The analysis focused on both the manifest and latent content and started with reading the transcripts from the interviews to gain familiarity with the content. K.H. and C.S. were mainly responsible for the analysis. NVivo r1/20.7.2 was used to extract 329 meaning units related to the aim from the units of each interview. The meaning units were condensed and labeled as 122 codes. In the analysis, the codes were compared and clustered into six subthemes and three themes, as described by Graneheim and Lundman [24]. A main theme, the latent content, is a theme describing the phenomena. Examples from the data analysis are presented in Table 2. This descriptive study took a qualitative approach that conformed to the Consolidated Criteria for Reporting Qualitative Research [25].

### 2.5. Ethics

This study was approved by the Swedish Ethical Review Authority (Dnr: 2023-04059-01). This study complied with the Declaration of Helsinki, and all of the informants were informed about and consented to participate in this study.

## 3. Results

### 3.1. The Main Theme

The main theme, “Healthcare professionals’ openness to opportunities for this technology for elderly patients”, is illustrated by the subthemes and themes elderly patients, virtual reality, and sustainable healthcare presented in Table 3, which describes the participants’ perspectives on the opportunities and challenges related to the application of immersive VR technology in patients aged 65 and older undergoing surgery with regional anesthesia and sedation.

### 3.2. Elderly Patients

This theme describes attitudes about elderly patients in relation to the use of distraction with VR.

#### 3.2.1. VR for the 65+ Patient Group

This subtheme explores perspectives on the older generation, specifically regarding their familiarity with technological advancements. The elderly were perceived to have limited knowledge of such developments, which may mean that they lack preconceived notions about VR. For instance, many may not even recognize a pair of VR glasses. This raised critical questions: are older adults open-minded enough to embrace this technology, and will it be effective for them?

A general attitude emerged where it was perceived that since the elderly do not keep up with technological developments, they may find VR unpleasant and difficult. But it was also believed that VR was an opportunity for them to try out the technology and that the experience could be positive. The perception was that older people often have difficulties with the settings and buttons on equipment. But the VR glasses were ready to use, so the participants thought that they may be able to use them effectively.

“This is a generation that may not really be running around with VR glasses at home, so it could be an interesting opportunity to try it out [5].”

The participants thought that older patients have less need for distraction in general because they have lived a long life and are calmer than younger patients. The elderly also experience age-related changes, such as poor vision and hearing, that need to be considered when VR is used. The perception was that the VR glasses as they were shown during the interview were compatible with glasses and hearing aids. The perception was that hearing aids would work well with the headphones used for music in this study, so it should not be a problem. Certain questions did arise, such as whether it would be possible to adapt or adjust the distraction for those with poor vision so that they could achieve the full experience of distraction. The perception was that discomfort would be dependent on how long the patient used the VR glasses. The ability to use VR was also assumed to be affected by the patient’s position during surgery.

The suggestion was that distraction with VR could be used throughout the entire perioperative care process for the elderly population. The participants had previous experience using music and radio through headphones or a speaker in the room as a distraction. The preoperative use of VR in the waiting room could offer an opportunity to immediately comfort anxious patients.

“When you are sitting there waiting, perhaps in the waiting room or pre-operatively, because it is difficult for the patient to sit and wait for the operation and it takes time. It might work well [7].”

The view was that there should not be too many different choices for distraction because this could result in stress for patients who are already in a vulnerable situation. It is assumed that what elderly patients would like to view through VR varies significantly, from nature, sea, travel, or something else calm and quiet to a beach, waves, a forest, or something that inspires meditation. Others might want to watch football. Those who are hypervigilant may not want to watch something calm. The distraction should perhaps have an antecedent, a common thread, where the patient experiences something they have not experienced before. This could be something soothing, a little predictable but enjoyable. Open landscapes are a good option for counteracting claustrophobia.

“If you want to actively engage in something or if you want to have something restful [17].”

It was suggested that VR could also be used postoperatively. Moreover, maybe patients who have used VR and come back for an examination or a treatment may feel that the distraction works well, and VR could make it easier for them at the next visit.

#### 3.2.2. Motivate and Inform the Patient

The participants also pointed out their role in motivating elderly patients to use VR. They are responsible for ensuring person-centered care and using patients’ resources. By telling elderly patients about the benefits of using VR, they can offer a distraction that shields them from the environment and contribute to a positive experience and perhaps contributes to a faster postoperative recovery.

“If we roll into the operating theater with a patient and there is so much about the room that you see and then you throw on a pair of VR glasses, it might be… then maybe the threshold is lower for ‘No, this … I don’t want to be involved, I want to stop this now’, but if you have seen them before, ‘Yes, exactly, those were the ones I have seen…’, they know how they feel [5].”

The participants stated that elderly patients need information in advance before using VR. This could include preparations at the preoperative visit with the anesthesiologist, as well as a brochure being sent to their homes with information about the option to use VR as a distraction. The brochure should include pictures of the VR glasses. This would give them time to reflect on this option. In addition, patients could be given the opportunity to try the VR glasses before surgery.

Another point that the participants emphasized was that the information provided should be short and concise. The information should be individualized, and it would be important to inform them about the risk of nausea and how it should be handled if it occurs. If the patient can try the VR glasses beforehand, the risk of nausea can be detected.

The participants pointed out the importance of creating security for the patient. They said that they could talk to a patient while they used the VR glasses and that they would always remain by their side as usual. The patients can take off the VR glasses at any time.

“I think it is important to say that they can stop whenever they want. That they can communicate with us, that we will never leave them with these on and then walk away. Instead, someone is there who maintains contact with you and that you can stop at any time [5].”

One participant expressed that it could be a good idea to start to use VR with calm patients who already felt secure in order to create a foundation.

### 3.3. Virtual Reality

This theme describes opportunities for the use of VR and the challenges and risks associated with using VR to distract patients.

#### 3.3.1. VR as a Distraction

In this subtheme, the participants described how they believed they could use VR technology and the effects they expected for their patients. Distraction and diversion are fundamental in reducing patient stress. The perception is that the patient is in a vulnerable situation where without the prerequisites for distraction and diversion, their only focus is on the present situation and feelings of boredom. Elderly patients often have a history of pain problems, and the operating position and discomfort caused by the operating table often result in pain for the patient. The participants expressed that VR could be a good distraction from discomfort and pain or a complement to regional anesthesia and sedation.

“Many people think it’s fantastic to have the spinal, but they don’t want to know or hear anything about what is happening otherwise. Then they have nothing against being awake. But then it could help them to zone out from the room they are actually in [5].”

However, this approach is not likely to be suitable for all patients. The participants used headphones with music or the radio for distraction and believed that VR could be an additional option and an opportunity to improve the distraction further by having the patient use multiple senses. It could be an opportunity to offer the patient access to an “imaginary world”. If patients are immersed in a VR environment, it could distract them from their concerns, and even very anxious patients could benefit from the technology. It was also important that VR offers an opportunity to divert the patient from their fear of pain. Worries about pain from the procedure increase a patient’s anxiety level. The quality of the experience was considered crucial to the patient’s satisfaction with VR, where the level of distraction can impact discomfort, anxiety, and pain.

“But I think you have to get a little inside the experience, I think, in order for you to be diverted. You shouldn’t have room to think too much, I think [3].”

The participants stated that if the patient enjoyed the distraction and the images eased their anxiety, the effect would be beneficial. Potentially, VR could be effective enough for sedation to become unnecessary.

#### 3.3.2. Challenges and Risks with Using VR

To some extent, the participants related the challenges and risks of VR to their own experiences, where some had experience using VR, and others did not. They assumed that discomfort such as nausea, dizziness, and vomiting could be a concern for patients. Also, there is a risk that these discomforts may be exacerbated if sedation and VR are combined. Moreover, a group that is likely to have more difficulties using VR is patients with cognitive impairments and dementia.

“For people who get nauseous easily, perhaps it might not be so appropriate [1].”

Another obstacle could be if the patient feels uncomfortable with the hardware and software in general, as this could increase anxiety in the perioperative context. The hardware can also a present a risk of damage to the operating table. The participants also suggested that surgery could be an obstacle to the use of VR. They stated that there could be a risk of pressure ulcers and the patient could become so deeply distracted that they would not feel the pressure. The loss of control for the patient could be problematic. This could be especially difficult for patients with a strong desire to maintain control. Feelings of being trapped and claustrophobia could arise in some patients. For surgery in the head/neck region, draping is used closely around the patient’s face. VR equipment could then lead to feelings of confinement. There is also the risk that the VR equipment could reduce the space for surgery. Another unwanted effect of using VR would be it contributing to feelings of exclusion and non-participation among patients. The VR experience itself could be uncomfortable if it was seen as too realistic.

VR could also mean a loss of control for healthcare professionals. If a patient is highly distracted by VR, it becomes difficult to communicate with the patient. This could be problematic when patient participation is needed or if another action needs to be performed, such as a new IV tube.

“The patient disappears into their own world and is not really involved. But they still have to be able to hear what we say. We have to be able to communicate with them when they need to hold their breath or say, ‘Now it is very important that you lie completely still’ [16].”

Another risk could be that VR affects a patient’s experience of pain. The nurse’s pain assessment could then be affected. Some nurses would prefer to be able to interrupt VR if they needed to give the patient pain-relieving drugs, while others see no obstacle in providing pain relief while maintaining the ongoing distraction through VR. The most frequently mentioned risk with the use of VR is that the patient may not lie still or could move unexpectedly.

There was also concern that the VR equipment could malfunction or interfere with other equipment used in the operating room. This would be a risk to patient safety and have an impact on a tight schedule. The use of the VR equipment must not cause technical problems to the extent that a nurse loses focus in monitoring a patient’s vital signs. Questions were raised regarding who would be responsible if the VR equipment did not work. Training would be needed if this responsibility were assigned to nurses. To sum up, the view was that it would be important for the equipment to work well and not malfunction, which would otherwise lead to irritation. The participants raised questions about cleaning and how the equipment should be handled. Existing equipment is cleaned with alcohol, so the perception was that this should not be a problem if care was used with the glasses at all angles.

“Where it makes contact with the face, there is this soft part and that’s the question and… and I don’t know how the glass… like what can we use on… what can we spray there [6].”

The participants pointed out that the glasses must be able to be cleaned in a safe, efficient, and simple way that is not too time-consuming and without the risk of alcohol damaging the lens.

### 3.4. Sustainable Healthcare

This theme describes how the participants thought that the use of VR as a distraction could promote sustainable healthcare that delivers high-quality care in an affordable way while minimizing the environmental impact.

#### 3.4.1. Balancing the Patient’s Sedation

The participants perceived sedating elderly patients to be a challenge. They are often fragile and sensitive to drugs, which poses a risk of unwanted effects from sedation. Distraction is needed, and if VR could be used, the participants noted several advantages.

“Yes, it is. It’s a bit of a difficult balance, where if you give too little, they might be anxious, fidgety or feel a bit of pain. And if you give too much, their breathing drops off, and then it may be that there is an anesthetic that you might want to avoid for various reasons [10].”

Using VR as a distraction could mean that the risk of respiratory depression is reduced. The negative impact of sedation on cognition and its hemodynamic effects can also be avoided, as can the risk that the patient will become unconscious and have their motor function affected by sedation. The participants expressed that VR may even make it possible to reduce the dose of opioids when they are used in addition to sedative drugs.

“The less you take in that is… what can you say, unnatural, the better it is, I think [3].”

The participants pointed out that perioperatively, the nurse needs to communicate with the patient, which would be facilitated if VR use reduced the amount of sedation needed. When using sedation, it is difficult to reach a suitable level where the nurse can communicate with the patient. Moreover, by reducing the use of drugs and using VR as a complement, it may also be possible to increase patient comfort. To sum up, the perception was that overall, patient safety would increase with the reduced use of sedative drugs.

#### 3.4.2. Economic and Environmental Impacts

Overall, the participants see several economic benefits of using VR. First, there is a reduction in costs if the use of drugs can be decreased. The waste of anesthetics such as propofol will also be an important cost factor. The reduced use of drugs would also be positive for the environment due to reduced demands for the manufacture of drugs and the patient’s excretion of drugs. The reduced consumption of disposable materials and the reduced need to administer oxygen to the patient would contribute further to a lower ecological footprint.

However, the participants also had doubts about the economic benefits of VR. The sedative drugs used have such a short half-life that using VR would not affect the patient’s time to recovery. The participants also mentioned that the VR equipment could break. If it is expensive to repair the equipment, it will not result in significant cost savings because the sedative drugs are not particularly expensive.

Some participants expressed an opposing point of view, stating that even if the VR glasses are expensive, they are reusable and therefore likely to be a more cost-effective alternative. In addition to the fact that the cost of VR is higher than that of alternative distractions (music and radio), it entails the introduction of something new in addition to what is currently available.

Another view expressed was that patient well-being could be improved, as VR use may generate a shorter course of care among patients. It can also have a positive effect on patients who return. If patients have a positive experience and feel secure using VR, there is a reduced risk that the patient will request anesthesia on the next occasion, which is a more expensive option.

The amount of time available for patient care could also be affected if the patient’s recovery from surgery was faster and the time needed for a procedure or surgery was reduced. From the participants’ point of view, VR use would mean that patients would tend to go home directly after a procedure/surgery and would be able to drive themselves home. More patients could therefore have access to care. And by reducing the amount of time in the healthcare facility, the patient has a lower risk of contracting hospital-acquired infections.

“[We] can give a little less sedation, but it’s not really that expensive, so I don’t really know. I think it’s more about an experience… that is, a… maybe slightly easier on the budget [17].”

## 4. Discussion

This study reveals a certain degree of ageism in healthcare, but most of the participants have a positive attitude towards the use of VR immersion in elderly patients. The participants reported that they are currently engaged in developing proposals for adaptations and new approaches that will help elderly patients. Perceptions about VR were founded in attitudes and previous experience—or a lack of experience—using VR. Shiner [26] showed that care providers tend to have a positive attitude towards VR and suggested that it be used in clinical activities, regardless of previous experience with VR, but also highlighted that the effectiveness of VR use is dependent on healthcare professionals’ conditions, as well as their knowledge and perceived capability [26]. This is consistent with studies that found barriers for VR use, as well as insufficient technical knowledge or skills [27,28]. In a review, Flynn [29] explored the use of VR among older adults living with dementia and emphasized the need for further research that included older adults living with dementia, as well as key stakeholders in the design process. At present, drugs and non-pharmacological measures (e.g., music) are used to reduce patients’ anxiety and fear. The general view is that distraction with VR could also be beneficial and an additional option used to address the challenge of balancing the need for sedation. The BIS could be used to reduce the dose of sedative drugs [30]. Despite this, assessments of a patient’s sedation level are often subjective and indirectly monitored through cardiovascular and respiratory function or based on a patient’s verbal response or response to tactile stimuli [31]. Based on their professional roles and perspectives, the participants in the current study suggested areas for the use of VR technology throughout the whole perioperative course. In addition, they identified risks, obstacles, and potential effects on patient safety with the use of VR. The assumption was that using VR as an additional distraction could have a positive impact on sustainable development and contribute to economic gains. In healthcare, the operating theater is one of the three largest contributors to greenhouse gas emissions [32]. In the context of this study, oxygen delivery plays a significant role in the carbon footprint of healthcare during spinal anesthesia [33]. Depending on the circumstances, distraction with VR during surgery could reduce the need to administer oxygen. The impact of propofol on the environment is also indirect, as it relates to the need for electricity for the syringe pump, the drug manufacturing process, packaging and transportation, the consumption of single-use plastics, and drug wastage and disposal [34]. Drug waste has an impact on the environment and affects costs [34,35]. The healthcare sector has lagged behind other sectors in terms of social responsibility and environmental sustainability. Healthcare professionals have a crucial role in advocacy and change management, both as individuals and at an organizational level [32]. When it comes to financial evaluations, it is important to take a broader perspective. In one review, Gómez [36] shows that when performing health-related economic evaluations, such as evaluations of virtual reality interventions, the patient’s perspective and the impact of VR are inextricable parts of the health economic evaluation. It is evident that the participants’ perspective in this study is valuable for the development and implementation of VR. For the implementation of VR distractions in the perioperative care environment, Rogers’ theory of the diffusion of an innovation in a clinical setting can be used. As mentioned above, Rogers [20] found that characteristics such as an individual’s ability to learn to use the innovation, the observed effects of the innovation itself, the motivation and ability of users, and the assessment of the implications in an organization influence whether an innovation is adopted. Barriers to the use of VR in healthcare include the maintenance of the equipment, the system, and the provider, including the associated costs; insufficient IT support; a lack of technical knowledge and skills; and time pressures [26]. This is in line with the present study, where the participants expressed concern about the maintenance and cleaning of the equipment and who would be responsible for the equipment, especially if it malfunctioned. The tight schedule places high demands on well-functioning routines in the event of technical problems with the equipment. The process of deciding on whether an innovation will be adopted or not goes through five stages: knowledge, persuasion, decision, implementation, and confirmation [20]. The implementation of perioperative VR distractions presents a challenge, and routines and behaviors will need to change. Behavioral science uses a strategy known as “nudging”, which is a strategy used to persuade people to behave in the desired way by changing their choices but without barring or removing any options [37]. If there are positive outcomes for patients aged 65 and over when using VR distraction, nudging could be used as a strategy to encourage healthcare professionals to choose VR over sedation and to ensure that patients feel comfortable trying VR. Still, influencing behavior also comes with great responsibility. An ethical approach is important when it comes to nudging. In a scoping review by Kouijzer [38] on the virtual reality implementation process in various healthcare settings, the author highlights the importance of involvement in the entire process, not just in identifying barriers. A successful implementation process needs to be supported by implementation frameworks and should preferably focus on behavioral changes among stakeholders, such as healthcare staff, patients, and managers.

The aim of this study was to investigate the opportunities and challenges related to the application of immersive virtual reality (VR) technology in patients aged 65 and older undergoing surgery with regional anesthesia and sedation. The qualitative content analysis was a suitable method for investigating the perspectives of healthcare professionals in different disciplines and with different seniority levels. To boost this study’s credibility, dependability, confirmability, and transferability, we rigorously followed all of the steps described by Granheim and Lundman in the analysis [24]. The sampling method aimed to include professionals from different disciplines with different durations of experience from three different surgical departments. This variation strengthens the credibility of this study since phenomena are elucidated from different perspectives [24]. Since there is a rapid pace of technological development in society, this study’s dependability is strengthened by the fact that the interviews were performed in the same way by using an interview guide and relying on a short data collection period [24]. In addition, the VR glasses were available during the interviews to enhance knowledge of the phenomena. The researchers’ preunderstanding has an impact on a study’s confirmability. This was considered in this study but had to be assessed in relation to the research question, which dealt with new technology and new working methods. However, the interviews were conducted by individuals with experience in this context, which was a prerequisite in this context for understanding the participants’ perceptions of the phenomena under study. The interviews were transcribed by a professional transcriber, and NVivo r1/20.7.2 was used in the analysis. The analysis thus became more transparent for all researchers, which strengthened its confirmability. This was strengthened by the authors’ discussions of the findings during the analysis [24]. To facilitate readers’ assessment of the transferability of this study, a clear description of the interviews and an analysis of the results are presented and illustrated with quotations [24].

## 5. Conclusions

The healthcare professionals in this study were generally positive towards the use of VR in elderly patients, highlighting both opportunities and potential barriers or risks. A distinction emerges in their perceptions about the opportunities and risks. The software and its content appear to be more closely associated with opportunities, while the hardware—the VR headset itself—is seen as a potential risk, particularly due to the difficulty communicating with patients when they are equipped with immersive goggles and headphones. The participants indicated that modifying their working methods in perioperative environments with high workloads presents a significant challenge. Successful implementation requires the active involvement of elderly patients and key stakeholders in the design and integration processes. Furthermore, a solid foundation based on training, clearly defined responsibilities, and strong IT support is essential. The implementation process may also be reinforced through subtle nudging strategies; for instance, healthcare professionals could wear pins with the message “Ask me about VR distraction as an alternative to medication” to foster awareness. In the long term, participants anticipated positive impacts of VR on healthcare environments and costs, potentially expanding the access to care for a larger patient population.

## Figures and Tables

**Table 1 healthcare-13-00669-t001:** Sociodemographic and professional characteristics of participants (n = 17).

	Mean	Range
Age, years	46	33–63
Number of years of healthcare	22.5	9–46
experience		
Profession		
Certified registered nurse anesthetist	9	
Operating room nurse	3	
Radiographer	3	
Critical care nurse	2	
Gender, M/F	5/12	

**Table 2 healthcare-13-00669-t002:** Examples from the data analysis.

Meaning Unit	Condensed Meaning Unit	Code	Subtheme	Themes	Main Theme
Because the study is for those 65 years and older, and I don’t know how to approach it for that age group because I think that……of it to work, maybe you need to be kind of open to it, receptive in any way.	Study for the elderly, works if you are receptive	Older people need to be receptive	VR for the 65+ patient group	Elderly patients	Healthcare professionals’ openness to opportunities for this technology for elderly patients
So information and things like that, above all, generally provide a great sense of security and then you’re half way there. As I say, they feel secure and trust, and so on, before they come here, and know that they might have to use a pair [of VR goggles] then you’ve done some of that part, as I have mentioned, when you greet the patient and they feel secure, and that you check on them later.	Information provides a great sense of security, they are halfway there when they arrive and know that they will use a pair of goggles	Information about VR before	Motivate and inform the patient		
But think that maybe they can be used precisely to make the patient zone out a bit and kind of shift to other thoughts and not maybe lay down and not be so nervous and tense up and think it’s hard to be …in surgery. I guess that’s more what I’m thinking, as a… also a way to relax for eldery patients.	Used to make the patient zone out, shift their thoughts, and not be nervous, a way to relax	Distracting them with VR	VR as distraction	Virtual reality	
Yes, there is one more aspect. Where some people easily feel ill easily, it might not be so appropriate.	Not suitable for people who are easily nauseated	Easily nauseated	Challenges and risks with using VR		
But we have some who are… they are so senstive to this… you have…you give a little pain relief and then nothing happens, and then you give another little more, and then the full effect of everything comes, and so the breathing slows down and saturation is poor.	Sensitive to pain relief; nothing happens at first, then the full effect is felt, and saturation is poor	Maintaining spontaneous breathing	Balancing the patient’s sedation	Sustainable healthcare	
But post-operative costs, that the patient goes home faster, gets out of the surgical ward faster, if you don’t have to give them as much sedation. Then, it certainly costs them quite a bit. But if you buy in and have something like that, and in the long run, it can certainly pay off.	The patient returns home from the surgical ward more quickly and does not need a lot of sedation; there are costs when you buy in the technology, but it can pay off.	More sedation leads to longer care	Economic and environmental impacts		

**Table 3 healthcare-13-00669-t003:** Overview of the results.

Main Theme	
Healthcare professionals’ openness to opportunities for this technology for elderly patients	
Themes	Subthemes
Elderly patients	VR for the 65+ patient group
	Motivate and inform the patient
Virtual reality	VR as a distraction
	Challenges and risks with using VR
Sustainable healthcare	Balancing the patient’s sedation
	Economic and environmental impacts

## Data Availability

The qualitative data based upon which this analysis was conducted are not publicly available due to ethical concerns regarding the confidentiality of the participants. Further, consent was not obtained from the participants to share information from the interview transcripts with third parties not involved in this research, and the ethical approval for this study does not permit such information being shared.

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
