# Peer review of "Healthcare Professionals’ Perceptions of Pre-, Peri-, and Postoperative Virtual Reality Immersion in Elderly Patients"

_healthcare, 2025, doi:10.3390/healthcare13060669_

Round 1

Reviewer 1 Report

Comments and Suggestions for Authors

It is an innovative work, and for me, it is truly attractive. It deals with a subject that has great expectations for the future. 
I would encourage the authors, however, to make some modifications to the manuscript. 
1.- The main objective should be a little more concrete: “to explore the possibilities...” is rather ambiguous and generalist. 
2.- I think that in the Results section explanations and justifications more typical of Discussion have been mixed. It would be convenient to change it. 
3.- In Discussion, I think that the first sentence deserves some explanation. At what ages was the study carried out?
I think that the economic valuation should be much more profound, not only the direct costs derived from pharmacology can be quantified, but we should also take into account the changes in hospital stays.
On the other hand, I think there should be a specific section dedicated to the risks derived from the use of these technologies in the operating room. 

Reviewer 2 Report

Comments and Suggestions for Authors

In this qualitative study the authors evaluate the perceptions of healthcare professionals towards using immersive VR technology in patients over 65 years of age that are due to undergo surgery with regional anesthesia plus sedation. The authors theoretisize that the use of VR will lead to less sedation and pharmacological intervention and therefore reduce side effects.

The authors conducted semi-structured interviews with 17 healthcare professionals. While these professionals were from different areas of anesthesia, it is noted that no physicians were included. This would have increased the quality of the study and should be explained by the authors. Also why were radiographers included in the study, they are not directly involved in the primary care of the subjects of the the study? 3/17 is an important percentage. With this in mind, the recruitment process and inclusion/exclusion criteria for participants would benefit from additional detail. 

The remaining methodology conducted by the authors is appropriate for the study design. To summarize, participants viewed the use of VR as a distraction tool that would reduce anxiety and therefore enhance patient experience, and lead to reduction in drug usage and decreased healthcare cost. The discussion section is adequate.

The manuscript could include tables summarizing key themes and subthemes that would improve readability. 

Reviewer 3 Report

Comments and Suggestions for Authors

• Even if it seems to be more a science-fiction discussion than actual practice, these devices and VR have already entered our lives. Due to the novelty of this field, exploring the use of VR in the perioperative setting of elderly patients will be a subject of great interest in the years to come.

• What specific improvements should the authors consider regarding the methodology? What further controls should be considered?   The study participants (medical staff) should be previously instructed in using VR because any hesitation in offering convincing answers to patients when presenting the VR option can cause them to refuse this technique. The best option is for them to be enthusiasts of using VR as the best solution against stress. When discussing this subject, one should asses the pros and cons of using the VR for patients during surgical procedures. Summarizing: Pros: Some studies demonstrate that VR reduces the patients` anxiety and stress and also the need for pain medication Cons: Important - potential electromagnetic interference with surgical equipment and pacemakers - potential motion sickness and discomfort less important - high costs and technical obstacles for implementation in the OR - length of the learning curve for both medical professionals and elderly patients  - the need for user-friendly interfaces with simplified controls   Prerequisite challenges include educating elderly patients on VR for surgical stress management, overcoming the technological barriers to using digital technologies, and overcoming the psychological resistance to an unknown device that can be harmful. Overcoming the lack of awareness and misinformation is a crucial step, so doctors and hospitals should launch educational campaigns about the benefits of VR before using the technique, mainly among elderly patients.     • Are the references appropriate? Yes, the references are appropriate and recent; more than half are under five years old.    • Any additional comments on the tables and figures? No comment about the tables and figures
